# Electrochemical Biosensors Based on Convectively Assembled Colloidal Crystals

**DOI:** 10.3390/bios12070480

**Published:** 2022-06-30

**Authors:** Amane Shiohara, Christopher D. Easton, Beatriz Prieto-Simon, Nicolas H. Voelcker

**Affiliations:** 1Drug Delivery, Deposition, and Dynamics, Monash Institute of Pharmaceutical Sciences, Monash University, Parkville, Melbourne, VIC 3052, Australia; amane@protonintel.com; 2Commonwealth Scientific and Industrial Research Organisation (CSIRO), Clayton, Melbourne, VIC 3168, Australia; chris.easton@csiro.au; 3Melbourne Centre of Nanofabrication, Victorian Node of the Australian National Fabrication Facility, Clayton, Melbourne, VIC 3168, Australia; 4Department of Electronic Engineering, Universitat Rovira i Virgili, 43007 Tarragona, Spain; 5ICREA, Pg. Lluís Companys 23, 08010 Barcelona, Spain

**Keywords:** electrochemical biosensors, convective assembly, colloidal crystal, virus detection

## Abstract

Rapid, sensitive, selective and portable virus detection is in high demand globally. However, differentiating non-infectious viral particles from intact/infectious viruses is still a rarely satisfied sensing requirement. Using the negative space within monolayers of polystyrene (PS) spheres deposited directly on gold electrodes, we fabricated tuneable nanochannels decorated with target-selective bioreceptors that facilitate the size-selective detection of intact viruses. Detection occurred through selective nanochannel blockage of diffusion of a redox probe, [Fe(CN)_6_]^3/4−^, allowing a quantifiable change in the oxidation current before and after analyte binding to the bioreceptor immobilised on the spheres. Our model system involved partial surface passivation of the mono-assembled PS spheres, by silica glancing angle deposition, to confine bioreceptor immobilisation specifically to the channels and improve particle detection sensitivity. Virus detection was first optimised and modelled with biotinylated gold nanoparticles, recognised by streptavidin immobilised on the PS layer, reaching a low limit of detection of 37 particles/mL. Intact, label-free virus detection was demonstrated using MS2 bacteriophage (~23–28 nm), a marker of microbiological contamination, showing an excellent limit of detection of ~1.0 pfu/mL. Tuneable nanochannel geometries constructed directly on sensing electrodes offer label-free, sensitive, and cost-efficient point-of-care biosensing platforms that could be applied for a wide range of viruses.

## 1. Introduction

Early detection of pathogens is in high demand across a range of industry sectors, including environmental, food, agricultural, medical, and most urgently biosecurity [1,2,3,4]. Polymerase chain reaction (PCR) and enzyme-linked immunosorbent assay (ELISA)^5^ are conventionally used to detect pathogens [5]. However, these techniques suffer from certain drawbacks when there is a need to track a live infectious virus, irrespective of the presence of non-infective viral fragments [6,7,8]. Both PCR and ELISA are “back-to-base” techniques. Analysis requires potentially infective materials to be transported to a centralised laboratory, increasing the risk of transmission in the transport chain. Additionally, for both techniques, the time required for delivering time-sensitive results is days, compared to “on-site” or point-of-care sensors reporting within minutes. Electrochemical biosensors have been gaining attention due to their excellent sensitivity, low-cost production, user-friendly operation, short analysis time, and most of all, great potential to provide point-of-care detection [5,9]. In the drive to improve sensitivity and form factor, nanofabrication techniques have been developed to allow the sensing architecture to better match a biological target’s size (20–400 nm for viruses) [10]. Electrochemical sensing platforms when built up from nanostructures present a high surface area, which in turn, can result in remarkable enhancement of the sensitivity [11,12,13]. Morphological features, such as nanopores or nanochannels, can be site-specifically functionalised to capture an analyte. Upon recognition, nanochannels are partially blocked, which reduces the current passing through to the electrode, measured by a change in the electrode’s voltammetry [14]. This type of signal change offers a high dynamic range through signal amplification.

Nanochannel-based immunosensors are well suited for nanoparticulate analytes, which harbour viruses, as they selectively trap a chosen size while filtering out microscale components, cells, and cell debris [15,16,17,18]. Previous examples of nanochannel immunosensors—based on porous silicon and most commonly porous alumina structures [15,16,17,18]—have been fabricated to target serum proteins [16,17] and viruses [9] in complex mixtures. However, these examples rely on a highly fragile layered approach. Most examples use a fragile and brittle nanochannel membrane that is constructed separately from the electrode and subsequently combined prior to biofunctionalisation. An additional drawback to the above examples is the use of very hazardous etching (hydrofluoric acid or chromic acid) chemistries, which present difficulties in downstream manufacturing and the regulatory approval process—where patient, manufacturer or user safety is paramount. Here, we report a novel nanochannel immunosensor constructed directly on the sensing electrode for virus detection, which offers a streamlined fabrication technique that avoids hazardous etching steps. A densely packed monolayer of PS spheres provides tuneable nanochannels based on the negative-space between spheres, convectively assembled on the gold electrode. Depositing silica on this monolayer, using glancing angle deposition, minimises the area where bioreceptors are immobilised, so viral recognition occurs only on the nanochannel walls and not on the non-responsive top of the spheres. Taken together, our electrochemical immunosensing approach, exemplified via detection of intact MS2, could provide rapid, sensitive, selective, and portable virus detection for a range of viral targets.

## 2. Materials and Methods

### 2.1. Reagents and Instruments

4′ × 4′ gold-coated microscope slides (BF glass), coated with 12.5 nm chromium and 150 nm gold layers (Az1500 resist coated), were purchased from Telic Company (Santa Clarita, CA, USA) and diced into 1.5 × 2.5 cm^2^ pieces with a Disco DAD321 dicing saw. Carboxylated and non-carboxylated PS spheres with diameter of 350 nm were purchased from PolySciences Inc. Biotinylated gold nanoparticles (d = 40 nm) and streptavidin were purchased from Cytodiagnostics Inc., and Merck (Melbourne, Australia), respectively. The inoculum of MS2 bacteriophage and monoclonal rabbit anti-MS2 antibody were purchased from Tetracore Inc. (TC-7004-001, Rockville, MD, USA). Potassium ferrocyanide (K_4_[Fe(CN)_6_]), potassium ferricyanide (K_3_[Fe(CN)_6_]), *N*-hydroxysuccinimide (NHS), 1-ethyl-3-(3-dimethylaminopropyl)carbodiimide (EDC), sodium dodecyl sulphate (SDS), ethanolamine, phosphate buffered saline (PBS) tablets, 2-(N-morpholino)-ethanesulfonic acid (MES), and anti-goat IgG antibody (control) were purchased from Sigma-Aldrich (Melbourne, Australia). All reagents were prepared in Milli-Q water and used as received.

### 2.2. Preparation of Nanochannel-Based Sensing Electrode with Polystyrene Spheres

Nanochannels were created by a monolayer of PS spheres (d = 350 nm) convectively assembled on the surface of gold-coated microscope slides. 500 µL of the aqueous PS spheres solution was centrifugated at 14,000 rpm for 5 min and 400 µL of the supernatant was removed followed by the addition of 3 µL of 10% sodium dodecyl sulphate (SDS) solution. The gold-coated slides were sonicated in acetone for 5 min followed by another 5 min of sonication in ethanol. After rinsing with ethanol and drying under nitrogen, the slides were treated with UV-ozone using SAMCO UV-Ozone cleaner (Model CD240-10S2, Santa Clara, CA, USA) for 45 min. 5 µL of the PS spheres solution was deposited on the edge of the slide and convective assembly of the PS spheres was performed with the help of a C-863 Mercury DC Motor controller (Physik Instrumente (PI) GmbH & Co. KG, Karlsruhe, Germany). The PS spheres were immobilised on the gold surface by annealing the slides on a hot plate at 109 °C for 3 min. The surface was vigorously washed with Milli-Q water and dried under a stream of nitrogen gas.

### 2.3. Fabrication and Surface Modification of the Polystyrene Sphere-Based Biosensor

The immunosensor fabrication process and the surface modification of the nanochannels are illustrated in Figure 1. Silica was deposited on the top surface of the slide-mounted PS sphere monolayer with the glancing angle deposition technique by means of an E-beam evaporator (Intlvac Nanochrome II) with the following parameters: initial deposition rate of 0.5 Å/s, maximum power of 10%, slew rate of 0.3 %/s, final deposition thickness of 50 nm, no rotation, and no elevated temperature (~40 °C). The slides were tilted at 75° or 80° and deposition was performed with no rotation. The silica deposition was optimised first with non-carboxylated PS spheres, and the optimised parameters were applied to carboxylated PS spheres, the latter being used throughout the rest of the experiments. Afterward, slides were treated with UV-ozone for 1 min to increase the number of carboxylic groups on the spheres’ surface. The slides were mounted in a Teflon cell for immobilisation of bioreceptors, and electrochemical sensing was subsequently performed in situ within the cell. 0.2 M NHS and 0.1 M EDC solutions were prepared in 0.2 M MES buffer, pH 5.5. Both solutions were mixed in a 1:1 volume ratio, and 100 µL was incubated on the carboxylated PS sphere surface for 30 min at room temperature to form succinimidyl ester groups. The samples were washed with 0.01 M PBS for three times, and 100 µL of 125 µg/mL streptavidin or 100 µL of 25 µg/mL anti-MS2 antibody in 0.01 M PBS pH 7.4 was added to the cell and incubated overnight at 4 °C. Afterward the surface was washed with 0.01 M PBS three times and 100 µL of 0.1 M ethanolamine in 0.01 M PBS was added to the cell and incubated for 45 min at room temperature to quench the unreacted succinimidyl ester groups. The slides were washed again with 0.01 M PBS solution three times and subsequently used for analyte sensing. To demonstrate nanochannel blocking selectivity towards the biotinylated model particle or MS2 virus, a control was fabricated by incubating 100 µL of 125 µg/mL of anti-goat IgG antibody instead of streptavidin or anti-MS2 antibody on the surface of PS spheres following the same procedure.

### 2.4. Electrode Characterisation

The morphology of the immunosensor was characterised by means of scanning electron microscopy (SEM). The images of the PS spheres were taken after each fabrication step, annealing, silica deposition, and after gold nanoparticle detection. All the images were acquired on a FEI NovaNano SEM 430.

The attachment of the bioreceptors was characterised by X-ray photoelectron spectroscopy (XPS). XPS analysis was performed with an AXIS Nova spectrometer (Kratos Analytical Inc., Manchester, UK) using a standard protocol detailed elsewhere [19]. The following parameters were employed during analysis: X-ray source and power—monochromated Al Kα source at 180 W; system pressure—between 10^−9^ and 10^−8^ mbar; pass energy—160 eV (survey) and 20 eV; step size—0.5 eV (survey) and 0.1 eV (high resolution); emission angle—0° as measured from the surface normal; charge neutraliser—on.

Data processing was performed using CasaXPS processing software version 2.3.15 (Casa Software Ltd., Teignmouth, UK). The atomic concentrations of the detected elements were calculated using integral peak intensities and the sensitivity factors supplied by the manufacturer. Binding energies were referenced to the C 1s peak at 284.7 eV for aromatic hydrocarbon.

### 2.5. Electrochemical Detection

Solutions of biotinylated gold nanoparticles were prepared in 0.01 M PBS. The concentration for biotinylated gold nanoparticles was adjusted to 10^2^, 10^4^, 10^6^, 10^8^, and 10^10^ particles/mL. 100 µL of each solution was incubated on the sensor surface for 45 min at room temperature with mild agitation with a microplate shaker (Thermo Scientific Compact Digital Microplate Shaker) at 100 rpm. After each incubation, the electrodes were washed with 0.01 M PBS three times to remove unbound gold nanoparticles. Differential pulse voltammograms (DPV) prior to and after analyte incubation were acquired, monitoring the changes in the oxidation current intensity of a redox probe added into solution, reflecting the changes in diffusion through the nanochannels. All DPV measurements were conducted with the following parameters: increment potential of 0.004 V, amplitude of 0.05 V, pulse width of 0.2 s, sample width of 0.0167 s, pulse period of 0.5 s, scan rate of 0.008 V/s, quiet time of 2 s, and sensitivity of 10^−4^. For reference and counter electrodes, Ag/AgCl refillable electrode (eDAQ ET054-3, OD 6 mm, length 30 mm) and platinum wire (Sigma Aldrich 267201-400MG, 0.5 mm diameter 99.99%) were used, respectively. Each electrode was immersed in 850 µL of 2 mM K_4_[Fe(CN)_6_] and 2 mM K_3_[Fe(CN)_6_] mixture solution prepared in 0.01 M PBS, and the voltammograms were obtained by scanning the potential from 0 to 0.6 V. All electrochemical measurements were conducted with an electrochemical analyser (CH instruments, model 600D series, Austin, TX, USA) using a three electrode Teflon cell containing PS spheres on gold as the working electrode, silver/silver chloride reference electrode, and a platinum wire counter electrode. As a control measurement to validate the selectivity against, the target analyte (i.e., biotinylated gold nanoparticles) was incubated on the electrode with anti-goat IgG antibody immobilised on the PS spheres. Afterward, the streptavidin-modified monolayer was tested against non-target carboxyl-terminal PEG-coated gold nanoparticles, with similar size and shape to the target biotinylated gold nanoparticles.

For MS2 bacteriophage detection, MS2 solutions at concentration of 10^0^, 10^2^, 10^4^, 10^6^, and 10^8^ pfu/mL were prepared in PBS. The detection procedure, including incubation volume, time, and temperature, was conducted in the same manner as described above for biotinylated gold nanoparticle detection. In this experiment, as the first control, MS2 phage was incubated on the anti-goat IgG antibody immobilised monolayer. As a second control, the anti-MS2 antibody modified PS spheres monolayer was tested against non-target analyte, *ΦX174* phage.

The current intensity values were normalised as follows: *ΔI* = (*I*_0_ − *I*)/*I*_0_, where *ΔI* is the normalised current difference between *I*_0_, the initial current intensity prior to analyte incubation, and *I*, the current intensity after analyte incubation.

## 3. Results and Discussion

### 3.1. Characterisation of Silica Deposition on Polystyrene Spheres

The negative spaces between PS spheres create reproducible sub-micron-sized nanochannels when the spheres assemble into a close-packed monolayer and adhere via lateral attraction in the evaporative process of convective assembly. Sphere dimensions, therefore, dictate nanochannel diameter tuneable to target analyte detection. In order to reduce monolayer separation, the PS spheres were partially annealed to attach them on the gold surface with the optimised parameters (at 109 °C for 3 min). To confine bioreceptor immobilisation to the nanochannels—as opposed to the top of the spheres—glancing angle deposition was used to deposit silica on non-functional sensing areas on top of the spheres and any exposed gold surface area as a result of assembly defects, while preventing the deposition of silica on the lower-set nanochannels. Therefore, the gold surface at the bottom of these channels remains exposed. The coverage areas of the top of the spheres were compared by applying two glancing angles of 75° and 80°. Figure 1 shows SEM images of the PS sphere monolayer prior to and after silica deposition at 75° and 80° angles. The silica coated areas of 50 PS spheres of each angle were estimated by calculating the triangle shape of silica on the PS sphere with ImageJ software. At 75° the sample showed a larger silica coated area of 26.95 nm^2^ ± 4.1 nm^2^ on the top of the spheres compared to the one at 80° of 14.93 nm^2^ ± 2.76 nm^2^. The effect of sample rotation during the deposition was also tested, though no significant improvement was observed (Appendix A). The remaining experiments were hence conducted with PS sphere monolayers modified by silica deposited applying a glancing angle of 75°. This optimisation was carried out with the non-carboxylated PS spheres. However, to fabricate the biosensors, the optimised parameters for silica deposition were applied to the carboxylated PS spheres and a similar coverage of the top surface of the PS spheres was confirmed (data not shown).

### 3.2. Stepwise Characterisation of the Polystyrene Sphere-Based Biosensor

Preliminary experiments using carboxylated PS spheres with covalently bound streptavidin incubated with biotinylated gold nanoparticles did not result in a change of the oxidation current of the redox probe, indicating the absence of a significant blockage of the nanochannels. We suspected this is due to the limited amount of the functional group on the surface within the pores. As such, we investigated the effect of UV-ozone to introduce and increase the number of carboxylic groups on both carboxylated and non-carboxylated spheres. The UV-ozone treatment is not expected to change the surface functionalities on the silica-covered regions. XPS analysis was used to track the introduction of carboxylic groups and covalent immobilisation of bioreceptors (streptavidin or antibodies). First, the introduction of carboxylic groups was confirmed after UV-ozone treatment of non-carboxylated PS spheres [20,21] (Appendix A). The electrodes were exposed to UV-ozone for 0, 1, and 5 min, and the change of the elemental quantification, specifically O, and the relative fraction of component C5, assigned to O-C=O, fitted to the C 1s spectra were monitored. The introduction of carboxylic groups to the surface of the PS spheres was confirmed by an increase in O with UV-ozone treatment as well as an increase in intensity for component C5 in the C 1s spectra when compared to the untreated PS sphere monolayer. As UV-ozone treatment of PS is non-specific, a range of other carbon-oxygen groups were generated, as can be observed in the C 1s spectra. Next, the UV-ozone effect on commercially available carboxylated PS spheres was investigated (Appendix A). Results indicate that UV-ozone treatment increases the atomic concentration of carboxylic groups on commercially available carboxylated PS spheres, to a greater amount than observed on non-carboxylated PS spheres after 1 min of UV-ozone exposure (7.4 at.% vs. 6.3 at.%). From XPS analysis of commercially available carboxylated PS spheres without UV-ozone treatment, the total atomic concentration of carboxylic groups was 4.7 at.% (Appendix A). After 5 min exposure, carboxylated PS spheres showed an even higher atomic concentration of carboxylic groups (10.0 at.%) compared to non-carboxylated PS spheres (6.9 at.%), highlighting the advantage of using UV-ozone treated carboxylated PS spheres to obtain a high concentration of O-C=O groups on the surface. For the remainder of this study, to maximise a coverage of functional groups, we used UV-ozone treated carboxylated PS spheres. Afterward, the effect of UV-ozone treatment on the morphology of the PS spheres was characterised by SEM. The SEM images revealed the clear morphological change of the PS spheres after 5 min exposure, although no significant morphological change was observed after 1 min treatment (Appendix A) ^20,21^. As such, the bioreceptor immobilisation was conducted after 1 min of UV-ozone treatment.

After introduction of additional carboxylic groups on the carboxylated PS spheres’ surface, those groups were activated with NHS and EDC under mild acidic conditions to form amide bonds with the amine groups of the bioreceptor (either streptavidin or anti-MS2 antibody). The immobilisation of streptavidin was studied by monitoring the at.% of N and relative fraction of component C4 (N-C=O, O-C-O, C=O) (Appendix A and Figure 2 and Appendix A). An increase in both these values relative to the unmodified PS sphere monolayer confirmed the presence of streptavidin on the UV-ozone-treated PS sphere monolayer. A comparison between carboxylated PS spheres with and without UV-ozone treatment demonstrated a higher degree of EDC/NHS coupling and streptavidin loading was achieved for the UV-ozone treated samples (at.% of N 0.06 vs. 1.2—Appendix A). This result supports the decision to UV-ozone treat carboxlayed PS spheres to achieve a greater fraction of functional groups on the surface available for coupling.

### 3.3. Evaluation of Polystyrene Sphere-Based Biosensor Performance

To assess the sensing performance of the biosensor, biotinylated gold nanoparticles were used as a model system of intact virus detection. As shown in Figure 1, when biotinylated gold nanoparticles are incubated on the sensing electrode, they bind to streptavidin immobilised on the nanochannel walls, thus partially blocking the channels. The diffusion of the redox probe K_4_[Fe(CN)_6_] into the channels is hindered and thus the current intensity produced by its oxidation on the gold surface decreases [17,22,23]. The average diameter of the nanochannels was 96 ± 9 nm (calculated from 100 spheres) after silica deposition and UV-ozone treatment. Hence on the basis of their size, biotinylated gold nanoparticles with an average hydrodynamic diameter of 47 ± 13 nm (our model of intact virus) should be able to diffuse into the channels and bind to streptavidin immobilised on the channel walls [24,25]. For each concentration of nanoparticles, the oxidation current was measured prior to and after incubation. To confirm biotin-streptavidin binding each electrode was characterised by SEM after the sensing experiments. Control biosensors were prepared by immobilising anti-goat IgG on the surface of PS spheres, and then incubated with biotinylated gold nanoparticles (Figure 3E,G), and by challenging the streptavidin-modified electrode with PEG-COOH coated gold nanoparticles (Appendix A). Nanochannel confined sensing is shown in Figure 3A,C,E,G showing differential pulse voltammograms obtained upon incubation with different concentrations of gold nanoparticles in solution, as well as corresponding SEM images of the different samples with and without silica deposition (Figure 3B,D,F,H). Nanoparticles were observed on streptavidin-modified samples, with and without silica deposition, after incubation with biotinylated gold nanoparticles (Figure 3B,D). On the samples with silica deposition, differential pulse voltammograms obtained upon incubating gold nanoparticle solutions of increasing concentration showed a dramatic decrease of the oxidation current (by over 20 µA) with increasing gold nanoparticle concentrations (Figure 3A). Almost no gold nanoparticles were observed on the parts coated with silica (Figure 3B), indicating that the binding of gold nanoparticles mostly occurred in the nanochannel space instead of on the PS sphere surface. However, for the samples without silica deposition, the oxidation current measured did not show a significant decrease with an increase in nanoparticle concentrations (Figure 3C). One specific factor that could contribute to the reduced sensitivity in the non-silica coated sensors is the presence of defect lines introduced during the convective assembly process [26,27]. These defects reduce the sensitivity of the biosensor. Additionally, on the biosensor without silica deposition, some gold nanoparticles are observed on the top surface (comparatively larger) of the PS spheres, and those do not contribute to the change in electrochemical signals.

In contrast, for the control samples both with and without silica deposition, the oxidation current did not show a significant decrease across the concentration range (Figure 3E,G and Appendix A), and neither did the SEM images of these controls (Figure 3F,H and Appendix A) show the presence of any gold nanoparticles. The absence of gold nanoparticles in those SEM images indicates that nanochannel blockage occurs due to biotin-streptavidin binding. Concentration response curves for all electrodes with and without silica deposition are presented in Figure 4A. The sensitivity of the biosensor increased by 50-fold for the silica-deposited sample, which showed y = 0.2544x − 0.4 (R^2^ = 0.9623), compared to the non-silica-deposited sample, which showed y = 0.0053x + 0.0392 (R^2^ = 0.9298). The LOD of this biosensor was calculated with the concentration response curve and the equation: *Xb1 + 3Sb1* where *Xb1* is the mean concentration of the blank and *Sb1* is the standard deviation of the blank [9,28]. The silica deposited electrode shows a theoretical LOD of 37 nanoparticles/mL.

After successfully demonstrating the detection of biotinylated gold nanoparticles with the developed biosensor including the effect of glancing angle of silica deposition on the PS spheres’ surface, MS2 bacteriophage detection in PBS buffer was performed to show the potential of the platform as a virus biosensor. MS2 bacteriophage is an icosahedral, positive-sense single stranded RNA virus that infects *Escherichia coli* (*E. coli*) [29], the detection of this virus can be used as an indicator of faecal contamination in water [30]. The morphological similarity of this virus, including the diameter of around 30 nm, to the biotinylated gold nanoparticles (*d* = 47 nm) makes MS2 an ideal analyte to demonstrate the potential to apply this biosensor to intact virus detection. The electrode was modified with anti-MS2 antibody which has a diameter of ~10 nm, somewhat larger than the diameter of streptavidin (~2 nm). We also used two controls: one where the electrode was modified with anti-goat IgG antibody, and the other where the anti-MS2 antibody-modified electrode was tested against *ΦX174* phage, which has a similar size and morphology to MS2 [31]. As shown in Figure 4B, a gradual increase in normalised current intensity was observed in the biosensor with increasing virus concentration, while neither of the controls showed a significant change in the normalised current, indicating that this biosensor successfully detected MS2 bacteriophage in the PBS buffer. The biosensor calibration curve equation of y = 0.0187x + 0.0459 (R^2^ = 0.992) had a lower sensitivity compared to the biotinylated gold nanoparticle analyte with the silica coated PS spheres. This could be due to biofouling effect caused by some undesirable biomolecules including protein fragments of host bacteria *E. coli* that remained in the solution. The calculated theoretical LOD for the developed MS2 immunosensor is ~1.0 pfu/mL, which is lower than most of those reported previously, lower than the numbers observed in optical and electrochemical MS2 biosensors based on porous silicon transducers (6 pfu/mL^9^, 2 × 10^7^ pfu/mL [32], and 4.9 ± 0.8 pfu/mL [33], respectively), and lower than the numbers observed in porous alumina membranes (~7 pfu/mL) [34], and in sandwich-based immunosensors incorporating carbon nanotubes (9.3 pfu/mL) [35]. Partial passivation, with silica deposition, may have contributed to a new lower LOD for virus detection, as confinement of viral particle recognition to the nanochannels alone is absent on previously reported nanochannel-based sensors.

## 4. Conclusions

An innovative approach to detect whole viral particles with a nanochannel-based electrochemical biosensor was investigated with a monolayer of PS spheres directly assembled on a gold surface. A highly sensitive and selective immunosensor for intact virus detection has been fabricated using nanochannel architecture based on the negative space between close-packed PS spheres assembled on a gold electrode with the help of glancing angle deposition of silica on the top surface, confining viral recognition only within the channels. The spaces between the close-packed PS spheres were partially blocked upon the incubation of the biosensor with the target analyte. This partial blockage was monitored by observing the change in the oxidation current of a redox probe added into solution. The biosensor was successfully optimised via the detection of biotinylated gold nanoparticles, after immobilisation of streptavidin on the surface of PS spheres. Glancing angle silica deposition on the top surface of the PS spheres dramatically increased the biosensor sensitivity around 50-fold and lowered the LOD to 37 particles/mL in buffer for biotinylated gold nanoparticle detection. The application of the platform as a virus biosensor was successfully demonstrated by accurately quantifying MS2 bacteriophage, showing a calculated LOD of ~1.0 pfu/mL, lower than most previously published intact viral biosensors. This unique approach to produce nanochannel-based structures for electrochemical biosensors can reduce the cost, time, and the safety risk of biosensor fabrication. This platform exhibits the potential of point-of-care biosensing for a wide range of pathogens including viruses with short analysis time, low cost and label-free detection.

## Data Availability

Not applicable.

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
