# Peer review of "Electrochemical Biosensors Based on Convectively Assembled Colloidal Crystals"

_biosensors, 2022, doi:10.3390/bios12070480_

Round 1

Reviewer 1 Report

The manuscript deals with the electrochemical detection of MS2 virus, by means of an immunosensor fabricated onto PS/silica structures deposited on bulk electrode. The manuscript is well written, but some main issues should be addressed before the publication

Please add chapters in the result and discussion section to divide the several topics, in order to make the manuscript clearer.

Please add the complete used parameters for the DPV measurements in the material and methods chapter.

No Optimization steps have been performed, including the concentration of antibody anti-MS2 (not even reported in the manuscript), the incubating temperature and incubation time of the analytes. Please perform these studies before the calibration curve measurements.

What is the calibration curve equation of the analyte measurements (figure 4B)? Furthermore, add the standard deviation of the intercept and slope for the calibration curve equation in figure 4A. Please write in the main text

About the interference studies, does the ΦX175 phage have a similar size of the target virus? Furthermore, in order to have strong evidence of the selectivity of the sensor, please perform this study with at least 4 possible and reasonable interferences.

Reviewer 2 Report

The work is overall well organized and well written. The objective has been clearly stated and the conclusions are partially supported by the findings arising from the experimental part.

There are a few small points that should modify before acceptance:

- Line 105; "Scheme 1", it seems this scheme is missing in the manuscript. Please add the scheme to the manuscript.

- Line 149; "0.01M PBS", a space should be added after 0.01.

- Line 347; "0.1 M PBS", apparently it should be 0.01 M PBS.

Reviewer 3 Report

Shiohara et. al. report the use of nanochannel generated by silica deposited PS spheres for detection of MS2 bacteriophage. The electrode is well characterized and EC results does show an LOD of 1.0 pfu/mL. 

There are few aspects that might be better explained and few errors that needs to be clarified. 

Scheme 1 is missing.

Line 186/Figure 1- SEM image are subjective measure of silica coating area. It does look like Fig.1B has more silica coated area vs. Fig.1C. However, this is a visual observation and subject to observer bias. Is there a objective way to measure extent of silica coating like measuring diameters (or mean) of dia. of individual spheres after/before deposition (or deposition angle)? 

Line 230-"The SEM images 230 revealed the clear morphological change of the PS spheres after 5 min exposure, although 231 no significant morphological change was observed after 1min treatment"- This is interesting. What is the mechanism of morphological change observed? Is this been previously reported in literature?

Line 260-"average diameter of the nanochannels was 96 ± 9 nm after silica deposition and UV-ozone treatment"-Please mention number of PS sphere from which this mean diameter was calculated.

Line 269-"Control biosensors were prepared by immobilising anti-goat IgG on the surface of PS spheres, and then incubated with biotinylated gold nano-particles; and by challenging the streptavidin-modified electrode with PEG-COOH coated gold nanoparticles (Figure S8)"- I think this is not Figure S8. Figure S8 does not contain data on PS spheres with anti-goat IgG immobilized.

Few additional items that would be better described or included-

Line 75/Section 2.1 - It is suggested to provide catalog numbers where available to ensure reproducibility. 
Line 82-What is TC-7004-01? Is this catalog number of one the items obtained from Tetracore?
Line 91-"aqueous PS sphere solution" - Is the the same product mentioned in Section 2.1 (i.e. does the PS spheres come in soln.?)
Line 107-"E-beam evaporator"-What are the parameters of E-beam evaporation that determines amount of silica deposited? Please include other parameters that are relevant for repeatability. 
Line 112-"Teflon cell"-Please add this item in materials section with catalog number if available.
Line 148-"mild agitation"- Please describe what kind of agitation/if shaker was used, how long etc.
Line 155-Please describe relevant DPV parameters like scan rate or pulse size as relevant.
Line 157-Please mention Pt mesh or ref. electrode in materials incl. dia./length of wire counter electrode if made custom.

Round 2

Reviewer 3 Report

All the review comments have been incorporated.